# Preoperative Arterial Embolization of Musculoskeletal Tumors: A Tertiary Center Experience

**DOI:** 10.3390/cancers15092657

**Published:** 2023-05-08

**Authors:** Alice Kedra, Anthony Dohan, David Biau, Anissa Belbachir, Raphael Dautry, Alexandre Lucas, Mathilde Aissaoui, Antoine Feydy, Philippe Soyer, Maxime Barat

**Affiliations:** 1Department of Diagnostic and Interventional Imaging, Cochin Hospital, Assistance Publique-Hôpitaux de Paris, 75014 Paris, France; alicekedra@gmail.com (A.K.); philippe.soyer@aphp.fr (P.S.); 2Faculté de Médecine, Université Paris Cité, 75006 Paris, France; 3Department of Orthopedic Surgery, Cochin Hospital, Assistance Publique-Hôpitaux de Paris, 75014 Paris, France; 4Department of Anesthesiology, Cochin Hospital, Assistance Publique-Hôpitaux de Paris, 75014 Paris, France; 5Department of Musculoskeletal Imaging, Cochin Hospital, Assistance Publique-Hôpitaux de Paris, 75014 Paris, France

**Keywords:** blood loss, surgical, bone neoplasms, embolization, preoperative, musculoskeletal, gelatin sponge, microspheres, Onyx

## Abstract

**Simple Summary:**

Musculoskeletal tumors often require surgical treatment, which can result in substantial peri-operative blood loss. Preoperative transarterial embolization (TAE) is used to reduce peri-operative blood loss during the surgery of musculoskeletal tumors but there is no consensus about the actual place of TAE in the musculoskeletal tumor therapeutic algorithm, and there is no firm recommendation about its best technical approach. The purpose of this study was to report our experience in preoperative TAE of musculoskeletal tumors regarding the effectiveness of preoperative TAE in terms of blood loss and functional outcomes. For 31 patients, we found that TAE led to complete (58%) or near-complete (42%) tumor devascularization, allowing bloodless surgery in 71% of patients and moderate transfusion needs for the remaining 29%. A total of 27% of patients had complete improvement of the initial symptoms at the end of the follow-up, 15 (50%) with partially satisfying improvement, 4 (13%) with partially unsatisfying improvement and 3 (10%) with no improvement.

**Abstract:**

The purpose of this study was to report the effectiveness of preoperative transcatheter arterial embolization (TAE) of musculoskeletal tumors in terms of blood loss and functional outcomes. Patients who underwent preoperative TAE of hypervascular musculoskeletal tumors between January 2018 and December 2021 were retrospectively included. The patients’ characteristics, TAE procedure details, degree of post-TAE devascularization, surgical outcomes in terms of red blood cell transfusion and functional results were collected. The degree of devascularization was compared between patients who had peri-operative transfusion and those who did not. Thirty-one patients were included. The 31 TAE procedures led to complete (58%) or near-complete (42%) tumor devascularization. Twenty-two patients (71%) had no blood transfusion during surgery. Nine patients (29%) had a blood transfusion, with a median number of red blood cell packs of three (q1, 2; q3, 4; range: 1–4). Eight patients (27%) had complete improvement of the initial musculoskeletal symptoms at the end of the follow-up, 15 (50%) had partially satisfying improvement, 4 (13%) had partially unsatisfying improvement and 3 (10%) had no improvement. Our study suggests that preoperative TAE of hypervascular musculoskeletal tumors allowed for bloodless surgery in 71% of patients and minimal transfusion needs for the remaining 29%.

## 1. Introduction

Musculoskeletal tumors represent a heterogeneous group of benign and malignant conditions, including primary and metastatic tumors occurring within the skeleton, joint structures and muscles [1]. The most frequent condition is represented by bone metastases, whose incidence is increasing due to the prolonged survival of patients with cancers [1]. Primary musculoskeletal tumors are relatively rare, as they account for 0.2–0.5% of all malignancies in all ages [2].

All of these various musculoskeletal tumors may present as incidental findings, pain, loss of function or fractures [3]. As a result, they regularly require surgical or interventional radiology treatment to prevent or stabilize fractures and to improve the patient’s quality of life [4,5]. However, surgery can result in substantial peri-operative blood loss, leading to longer operative times, increasing peri-operative mortality and postoperative morbidity particularly because of allogenic transfusions [6,7]. Substantial blood loss occurs during the surgical resection of hypervascular musculoskeletal tumors, such as aneurysmal bone cysts, giant cell tumors, solitary fibrous tumors or osteosarcomas for primary musculoskeletal tumors, as well as bone metastases from renal cell and thyroid carcinomas [8].

Preoperative transarterial embolization (TAE) is often used to reduce peri-operative blood loss during surgery for musculoskeletal tumors [9]. Indeed, the goal of TAE is to occlude tumor vessels using embolic agents through a catheter that is selectively placed in an arterial branch feeding the tumor. Consequently, TAE devascularizes and maintains ischemia and necrosis in the center of the tumor. As a result, TAE may facilitate resection since the tumor often shrinks in response to ischemia, and thus, borders between the tumor and the surrounding tissues can become clearer [10]. However, there is no consensus about the actual place of TAE in the musculoskeletal tumor therapeutic algorithm, probably because of the diversity of this heterogeneous group of tumors. In this regard, one study revealed that preoperative TAE significantly reduced the mean estimated blood loss, allogenic transfusion volume and operative time when applied to bone metastases [11]. Similarly, TAE with cyanoacrylate demonstrated effectiveness for aneurysmal bone cysts and giant cell neoplasms, reducing morbidity and local recurrences [12,13]. In contrast, a recent systematic review found no evidence to support the use of preoperative TAE for bone metastases from thyroid cancer, lung carcinoma or prostate carcinoma [14].

Preoperative TAE of musculoskeletal tumors is often complex with non-negligible risk to adjacent structures. The current development of interventional radiology has led to an increasing choice of catheterizing devices, and since the first description of TAE using a gelatin sponge [15], many options for embolic agents are available, such as polyvinyl alcohol (PVA) particles, trisacryl microspheres, different liquid agents (N-2-butyl cyanoacrylate [NBCA], absolute alcohol, ethylene vinyl alcohol copolymer, and stainless-steel-fibered or platinum coils) [16]. To date, there are no firm recommendations in the literature about the best technical approach for the preoperative TAE of musculoskeletal tumors, and the predominant factor with respect to the choice of device or embolic material seems to be the experience and preference of the operator [17].

Therefore, the purpose of this study was to report our experience in preoperative TAE of musculoskeletal tumors with the wish of providing stronger evidence regarding the effectiveness of preoperative TAE in terms of blood loss and functional outcomes.

## 2. Materials and Methods

### 2.1. Study Population

Our institutional review board approved the retrospective data analysis (AAA-2021-08-037) and waived the need for written consent from the participants due to the retrospective design of the study. The database of our institution was retrospectively queried from January 2018 to December 2021 inclusive to identify all consecutive patients who had elective surgery for a musculoskeletal tumor after TAE. For all patients, preoperative TAE was decided during multidisciplinary musculoskeletal tumor board meetings when important intraoperative bleeding was anticipated due to the hypervascular presentation of the tumors. Inclusion criteria included (i) age > 18 years; (ii) history of a musculoskeletal tumor involving bone, joint or soft tissue that was benign or malignant and either primary or secondary; and (iii) musculoskeletal tumor treated using elective orthopedic surgery after preoperative TAE. Patients were excluded when no data regarding the TAE procedure or follow-up were available.

### 2.2. TAE Procedure Details

Each decision for surgical treatment with preoperative TAE was performed following the recommendation of a multidisciplinary musculoskeletal tumor board meeting. According to Ma et al., there are no imaging standard criteria for evaluating tumor vascularity since the correlation between the degree of enhancement on magnetic resonance imaging (MRI) and the vascularity degree found on angiography remains unclear [18,19,20]. As a result, pre-surgical TAE was considered for all musculoskeletal tumors assumed to be clinically or histologically hypervascular, regardless of the imaging features, or for musculoskeletal tumors whose imaging features were consistent with hypervascularization regardless of the clinical or histological characteristics.

All TAE procedures were performed in an angiography suite equipped with a C-arm fluoroscopic unit (Allura XperFD20, Philips, Amsterdam, The Netherlands) by a panel of five interventional radiologists (A.D., R.D., M.Bl, M.Bt., M.A.) with five or more years of experience in TAE under fluoroscopic guidance and strict aseptic conditions. Vascular access was obtained using the arterial, femoral or radial approach, depending on operator preference using the Seldinger technique. Introducer sheaths were used with variable lengths depending on the distance between the puncture point and the tumor.

The first step of TAE consisted of a careful analysis of the musculoskeletal tumor vascular anatomy. This was obtained via selective catheterization using a 4-French (F), 5-F or 6-F catheter of the main arterial branch(es) that was (were) supposed to vascularize the tumor. An angiogram was obtained to evaluate the number and the anatomical characteristics (ostium position, direction) of the feeding arteries (Figure 1). Particular attention was paid to the presence of extratumoral branches that arose from the feeders whose embolization might lead to ischemic complications. Then, each tumor-feeding artery was selectively catheterized using a microcatheter whose caliber ranged from 2.0-F to 2.8-F, depending on the tumor feeder’s size.

Tumor-feeding artery occlusion was undertaken using different embolic agents, whose selection was left at the discretion of the operator in the absence of robust recommendations. They included liquid embolic agents (NBCA (Glubran^®^, GEM Italy, Viareggio, Italy) or ethylene vinyl alcohol copolymer (Onyx^™^, Microtherapeutics, Irvine, CA, USA)), a gelatin sponge (Gelfoam^®^ (Pfizer, New York City, NY, USA)) or Embocube^®^ (Merit Medical, South Jourdan, UT, USA) in pieces of 2.5, 5 or 10 mm microspheres (Embospheres^®^, Merit Medical, South Jourdan, UT, USA) with various calibers (300–500 µm, 500–700 µm, 700–900 µm or 900–1200 µm). When the embolization of an extratumoral branch arising from a tumor-feeding artery could not be avoided and the ischemic risk was considered non-negligible, this extratumoral branch was also microcatheterized and occluded with metallic coils for protective purposes (i.e., “back door embolization” technique) [21].

At the end of the procedure, the final control angiogram was obtained to evaluate the degree of tumor embolization based on visual estimation of tumor blush reduction. TAE was considered complete when a reduction in tumor blush greater than 90% was observed, near-complete for a 75 to 90% reduction and incomplete for a reduction <75% [22]. Blush reduction was evaluated by the operator and indicated in the final report.

Hemostasis at the puncture site was performed by using manual compression or 6-F or 7-F closure devices, including Femoseal^®^ (St Jude. Medical, St. Paul, MN, USA), Angioseal^®^ (Terumo, Tokyo, Japan) or Starclose^®^ (Abbott Vascular, Redwood City, CA, USA).

All patients underwent orthopedic surgery the same day or the day after since it is generally recommended to perform surgery as early as possible after TAE, although no consensus exists regarding the time between TAE and further surgery [23]. The different surgical techniques employed were resection; curettage, which is associated with prosthesis, osteosynthesis or both; prosthesis alone and osteosynthesis alone. The decision regarding the surgical approach and technique was taken by a panel of several specialized orthopedic surgeons who reached a consensus opinion during multidisciplinary musculoskeletal tumor board meetings.

All patients were followed-up at regular intervals with clinical examination, in particular with an orthopedic consultation, during which functional results were evaluated (pain, mobility) with a minimal follow-up time of three months. Imaging examinations were also performed to search for local recurrence and distant metastases.

### 2.3. Data Collection

Medical, biological and surgical data were retrospectively recorded by the study coordinator (A.K., a fourth-year resident in radiology). They included (i) age at the time of surgery, (ii) gender, (iii) histological type of the tumor, (iv) localization of the treated tumor, (v) type of surgery, (vi) purpose of the surgery (therapeutic goal or symptomatic/functional goal in palliative surgical interventions), (vii) intraoperative blood loss, (viii) hemoglobin serum level before and after surgery, (ix) surgical complications if any, (x) local recurrence, (xi) global progression of the disease in case of malignant tumor and (xii) overall survival (OS).

In addition, the study coordinator recorded details about TAE including (i) the main arterial branch to the tumor, (ii) tumor-feeding arteries, (iii) embolized extratumoral branches, (iv) devices used (diameter and length of introducer sheaths, diameters of catheters, diameters of microcatheters, diameters of microwires, type of embolic agents and volume of iodinated contrast material in mL), (*v*) dose area product (DAP in mGy·cm^2^), (vi) air kerma (in mGy) and (vi) endovascular procedure-related complications. Main arterial branches were defined as large- or medium-caliber arteries that gave rise to several feeders vascularizing the tumor. Feeders were smaller arterial branches vascularizing the tumor in which embolic agents were injected.

### 2.4. Outcomes

For each patient, the difference in hemoglobin serum level (g/dL) between after and before surgery was calculated and considered as an indicator of peri-operative blood loss. The need for blood transfusions was estimated as the number of red blood cell packs received by the patient.

Functional results were evaluated according to the latest orthopedic consultation reports available during the follow-up. There was no standardized scale to evaluate functional results because of the heterogeneity of tumor localization and surgical interventions. Surgeons did not systematically employ quality-of-life questionnaires in their follow-up reports. However, they assessed the evolution of pain, mobility and consequences on daily life activities. All of these elements were retrospectively gathered and interpreted to define four levels of post-procedure evolution. A complete improvement was reported when the patient noticed a return to baseline autonomy without any residual pain related to the treated bone. Partial satisfying improvement corresponded to a significant but incomplete reduction of pain and disability, associated with a greater quality of life compared with the situation before surgery. When the patient noticed a slight improvement in pain and mobility with an important remaining disability, it corresponded to a partially unsatisfying improvement. Finally, lack of improvement was defined as the absence of significant improvement in pain, disability and consequences on quality of life. Local recurrence and global disease progression were noted for each patient.

### 2.5. Statistical Analysis

Statistical analysis was performed using version 4.1.0 of R software. Continuous variables were expressed as medians, interquartile ranges (q1; q3) and ranges [24]. Qualitative variables were expressed as raw numbers and proportions. The OS was estimated using the Kaplan–Meier method. The survival time was defined from the date of surgery to the date of death or the last follow-up visit. The percentage of complete or near-complete devascularization at the end of TAE was compared between patients who had peri-operative transfusion and those who did not using a chi^2^ test. The association between the degree of devascularization and the need for transfusion was evaluated by estimating the odds ratio and its 95% confidence interval. The level of significance was set at *p* < 0.05.

## 3. Results

### 3.1. Study Population

Thirty-three patients who underwent TAE for musculoskeletal tumors before elective orthopedic surgery were initially identified. Of these, two patients were further excluded because they were followed-up at another institution and no information about their functional status after TAE and surgery were available. The final study population included 31 patients (15 men, 16 women) with a median age of 67 years (q1, 55; q3, 74; range: 19–87 years). The initial characteristics of the 31 patients are summarized in Table 1. Among the 31 musculoskeletal tumors, 27 (27/31, 87%) were bone metastasis and mainly from clear cell renal cell carcinoma (17/31, 55%), and four (4/31, 13%) were primary musculoskeletal tumors (aneurysmal bone cyst, giant cell tumor, undifferentiated spindle and pleomorphic sarcoma, with one each). Tumors were mostly located in the thigh (12/31, 39%) or pelvic bones (6/31, 19%).

### 3.2. TAE Procedures

All TAEs were performed with morphine-based patient-controlled anesthesia (PCA). Other analgesic drugs, such as paracetamol, ketoprofen or nefopam, were added in four TAEs (30%), when morphine did not provide sufficient analgesia or when it was poorly tolerated during TAE. TAEs were performed under local anesthesia in 29/31 patients (94%); 1/31 patients (3%) had regional anesthesia and 1/31 patients (3%) had general anesthesia because he denied local anesthesia. A femoral approach was used in 30/31 (97%) patients and a radial approach in one patient (3%).

A total of 33 introducers were used for 31 TAEs, consisting of short introducers (20/33, 61%), with a length of 10 cm, or long introducers (13/33, 39%), with a length of 45 cm or 55 cm. The introducers’ calibers ranged from 4-F to 6-F. A total of 49 catheters were used for the 31 TAE procedures, with a single catheter used for the whole TAE procedure in 16 TAEs (57%). The catheter type was variable, depending on the anatomical configuration of the targeted artery, with calibers ranging from 4-F to 6-F. Two TAEs (7%) did not require the use of a microcatheter because the main feeding artery (posterior trunk of the intern iliac artery) was embolized with pieces of gelatin sponge (Gelfoam^®^). A single microcatheter was used in 24 TAEs (77%) and 2 or 3 in 5 TAEs (16%), with a caliber ranging from 2.0- to 2.8-F. A total of 36 microwires with calibers ranging from 0.014- to 0.021-inch were used since two microwires were needed for 7 TAEs (2%) and 3 for 2 TAEs (7%).

The embolic agents were metallic coils for 16 TAEs (52%), microspheres for 15 TAEs (48%), a gelatin sponge for 14 TAEs (42%), NBCA for 6 TAEs (19%) and Onyx^™^ 18 for 5 TAEs (16%). Some TAEs required the use of different embolic agents for the same tumor. The gelatin sponges were in the form of Gelfoam^®^ pieces (8/14, 57%) or Embocube^®^ particles that were 2.5 (2/14, 14%), 5 (3/14, 21%) or 10 mm (1/14, 71%) in size. The median number of coils per TAE was three (q1, 2; q3, 5.5; range: 1–10). The median volume of Onyx^™^ 18 was 1.5 mL (q1, 1; q3, 3; range: 1–5 mL). The Embospheres (Merit Medical, South Jourdan, UT, USA) had a caliber of 300–500 µm for two TAEs (13%), 500–700 µm for nine TAEs (60%), 700–900 µm for six TAEs (40%) and 900–1200 µm for one TAE (67%). Various calibers of microspheres could be used for a single TAE, depending on the result of the intercurrent angiographic controls in terms of devascularization. The Glubran^®^ (GEM Italy, Viareggio, Italy) dilution with ethiodized oil (Lipiodol^®^, Guerbet, Aulnay-sous-Bois, France) was 1:2 in two TAEs requiring the use of NBCA (33%), 1:3 in one TAE (17%), 1:4 in one TAE (17%) and 1:6 in one TAE (17%). The median volume of Glubran^®^ was 3 mL (q1, 0.4; q3, 3; range: 0.4–3 mL).

An arterial closure device, with a caliber of 6-F in 16/17 TAEs (94%) or 7-F in 1/17 TAEs (6%) was used in 17 TAEs (55%). The median amount of iodinated contrast material was 200 mL (q1, 100; q3, 200; range: 40–400 mL).

The median DAP was 62,117 mGy·cm^2^ (q1, 29,680; q3, 153,655; range: 29,680–587,656 mGy·cm^2^) and the median air kerma was 329 mGy (q1, 153.0; q3, 617.8; range: 67.1–3301.1 mGy).

The 31 musculoskeletal tumors were vascularized using a total of 48 main arteries. The tumor was vascularized by a single main artery in 14 TAEs (45%), by 2 main arteries in 14 TAEs (45%) and by 3 main arteries in 2 TAEs (7%). The main arteries from which the embolized tumoral feeders arose were the abdominal aorta, intern iliac artery, extern iliac artery, common femoral artery, deep femoral artery, superficial femoral artery, popliteal artery, axillary artery, humeral artery, ulnar artery and radial artery. A total of 84 tumor-feeding arteries were embolized during the 31 TAEs (Table 2). The median number of embolized feeders per TAE was three (q1, 2; q3, 3; range: 1–6). Extratumoral arterial branches were embolized in four TAEs (13%).

Regarding the final angiographic control, complete devascularization was observed in 18 TAEs (58%), and near-complete was observed in 13 TAEs (42%). No complications were reported in any of the 31 TAEs.

### 3.3. Surgical and Clinical Outcome

Among the 31 orthopedic surgical interventions, there were 16 resections (52%), 5 curettages (16%), 4 prosthesis placements alone (13%), and 6 osteosyntheses alone (19%). Resections were associated with prostheses in eight patients (26%), with osteosynthesis in two patients (7%) and with both in one patient (3%) (Table 3). Curettage was associated with a prosthesis for one patient (3%) and with osteosyntheses for two patients (7%). The surgical purpose was palliative for 27 patients (87%) with secondary tumors and curative for 4 with primary tumors (13%). Two peri-operative complications in two different patients (7%) were reported. One patient had diaphragmatic paralysis after general anesthesia, which required a short stay in the intensive care unit, but the patient recovered after a few hours and did not have sequelae. Another patient developed a surgical site infection during the month following surgery, which led to cardiac failure and death.

The median intraoperative blood loss was 300 mL (q1, 300; q3, 1000; range: 300–1000 mL). The median difference in hemoglobin serum level between after and before surgery was 1.9 mL (q1, 0.9; q3, 2.7: range: 0.1–6.2 mL). Nine patients (29%) required blood transfusions, with a median of three units (q1, 2; q3, 4: range: 1–4) of packed red blood cells. The nine patients who required blood transfusion underwent palliative orthopedic surgeries (7/9, 78%) for musculoskeletal tumors located in the sacrum (2/9, 22%), the acetabulum (2/9, 22%), the ilium (1/9, 11%), the femoral condyle (1/9, 11%), metaphysis (1/9, 11%) or the humerus (2/9, 22%). As a result, the location probably does not explain the need for blood transfusion. Nevertheless, the type of surgical intervention may explain the need for blood transfusion since 7/9 patients (78%) requiring blood transfusion underwent “complex” intervention, including at least two different techniques (i.e., the association of resection and prosthesis placement (6/9, 67%) or the association of resection and osteosynthesis (1/9, 11%)). Among the nine patients who required blood transfusion, four (44%) had near-complete devascularization on a control angiogram. Among the 22 patients who did not need any blood transfusion, nine (41%) had near-complete devascularization. The percentage of complete or near-complete devascularization at the end of TAE did not differ between patients who required blood transfusion and those who did not (*p* = 0.86). The degree of devascularization at the end of TAE was not significantly associated with the need for blood transfusion (odds ratio: 1.7, 95% confidence interval: 0.26–12.8, *p* = 0.70).

Functional results of the orthopedic surgical interventions, which were evaluated regularly from three months after surgery, could be evaluated only for 30 patients (97%) since one of them died 1.4 months after the surgery. Eight patients (27%) had complete improvement at the end of the follow-up, 15 (50%) had partially satisfying improvement, four (13%) had partially unsatisfying improvement and three (10%) had no improvement. Local recurrence was noticed for six patients (19%). Fifteen patients (48%) had global progression disease at the end of the follow-up. Five patients (16%) died at the end of the follow-up, yielding a median OS of 22.1 months (q1, 5.1; q3, 29.7; range: 1.4–40.7 months).

## 4. Discussion

In this study, we found that preoperative TAE resulted in a complete or near-complete devascularization of 58% and 42% musculoskeletal tumors, respectively. In addition, we found a peri-operative blood transfusion rate of 29%, with a median drop in hemoglobin serum level of 1.9 g/dL only after surgery. Our study suggested that preoperative TAE allows for bloodless surgery of hypervascular musculoskeletal tumors in 71% of patients and only moderate transfusion needs for the remaining 29%.

A preoperative TAE of a musculoskeletal tumor is a complex procedure. First, there is no strong recommendation about the optimal technique. As a result, the operator selects their approach depending on their own experience and preference and based on pre-TAE imaging findings. However, some situations are hard to anticipate and are ultimately identified intraoperatively. This is why for a single procedure, several catheters, microcatheters and embolic agents may be used. However, some elements may guide the decision, such as the vessel caliber and the presence of arteriovenous shunts or collaterals. A good knowledge of the properties of each embolic agent is also necessary to perform an effective and safe TAE. For example, we used liquid embolization agents, such as NBCA or Onyx^™^ 18, to provide fast and permanent occlusion. In fact, these embolic agents are known to induce more complete tumor necrosis than particulate agents [25]. Moreover, NBCA is mixed with ethiodized oil, making it visible during TAE. The best dilution ratio of NBCA with ethiodized oil seems to be 1:3 or less to avoid premature NBCA polymerization and ensure distal embolization. Nevertheless, NBCA and Onyx^™^ 18 are more difficult to manipulate and carry a greater risk of non-targeted embolization, which is an issue for these patients who must undergo potentially disruptive surgery. This is why the benefit–risk balance between optimal embolization in terms of devascularization and preservation of tissues adjacent to the tumor must therefore be carefully assessed. Particulate agents, such as PVA particles or tris-acryl gelatin, are easier to handle but their caliber must be carefully selected to avoid venous efflux, and those < 300 µm should not be used [17,26]. We do prefer tris-acryl gelatin particles because of their more regular shape compared with PVA particles, which decreases the risk of aggregation and catheter occlusion [27]. Gelatin sponge provides temporary occlusion and should be used when surgery is undertaken less than 15 days after a TAE [16]. Coils are also an option, particularly in the “back door embolization” technique, during which an extratumoral branch arising from a tumoral feeder is occluded for protective purposes to avoid ischemic complications. However, coils are not recommended as the sole embolic agent in tumor embolization because they cannot provide distal vessel embolization, which is necessary for hypervascular tumors with multiple collaterals [28]. The different embolic agents available do not provide the same effectiveness in terms of devascularization but they are complementary because of their different properties. We may recommend the association of an embolic agent providing distal embolization, such as NCBA, Onyx or tris-acryl gelatin particles, with the additional use of metallic coils to avoid non-target arterial occlusion.

In our study, a complete or near-complete devascularization was obtained in all tumors, regardless of the type of embolic agent. This made it difficult to establish recommendations, as no differences were observed in terms of the TAE efficacy between the different embolic agents. Our results with respect to devascularization were similar to those observed in the study by Kwon et al., who reported devascularization > 70% in 24 out of 25 patients (96%) with peripheral bone metastasis [29]. Although commonly used, visual estimation of devascularization may be subject to interobserver variability [30]. Cone-beam computed tomography (CBCT) may provide a more objective way to estimate tumor devascularization [31]. A comparison of tumor enhancement before and after TAE may be a more reliable method to analyze the degree of tumor devascularization. To overcome this limitation, in our study, control angiograms at the end of the TAE were performed with the catheter in the most proximal part of the main artery feeding the tumor with long-term fluoroscopic acquisitions to ensure that no arterial branches were left unembolized.

Our results in terms of blood loss should be interpreted on the basis of the low rate of blood transfusion required, which was consistent with the results of Kwon et al. [29]. We were not able to precisely evaluate the blood loss in mL because the information was rarely available in surgical reports, which may induce an interpretation bias. Moreover, our study did not include a control group of patients without preoperative TAE, and thus, our results should be interpreted with caution. We found no associations between the degree of tumor devascularization and the need for transfusion. This may be partly explained by the fact that all patients had a high degree of devascularization in our study. However, literature data reporting on the effectiveness of preoperative TAE of musculoskeletal tumors in terms of blood loss show conflicting results. In a randomized, controlled trial, Clausen et al. found no significant differences in blood loss estimated in mL and the need for transfusion between patients with and without preoperative TAE [32]. In contrast, Kato et al. found a significant difference in mean blood loss between patients who underwent TAE (520 mL; range, 140–1380 mL) and those who did not undergo TAE (1128 mL; range, 100–3260 mL) [33].

## 5. Study Limitations

Our study had some limitations. First, the retrospective collection of data about peri-operative blood loss was incomplete, with missing information regarding the exact volume of peri-operative blood loss. This study’s retrospective design did not allow for collecting objective and standardized information about the functional results of orthopedic surgeries. All of these elements may lead to an information bias. Second, we did not search for a potential association between the degree of tumor devascularization and patients’ quality of life after treatment nor local recurrence of the tumor because we did not have any control group. In addition, our population presented with great heterogeneity in terms of tumor localization, tumor type and surgical techniques, which made the subgroup analyses difficult.

## 6. Conclusions

In conclusion, the results of our study bring further evidence regarding the safety and efficacy of preoperative TAE of a musculoskeletal tumor. Preoperative TAE of a musculoskeletal tumor was found to be a safe procedure that allowed for bloodless surgery in 71% of patients and minimal transfusion needs for the remaining 29%. However, there is still a need for larger-scale, randomized and controlled studies to definitively confirm our results.

## Figures and Tables

**Figure 1 cancers-15-02657-f001:**
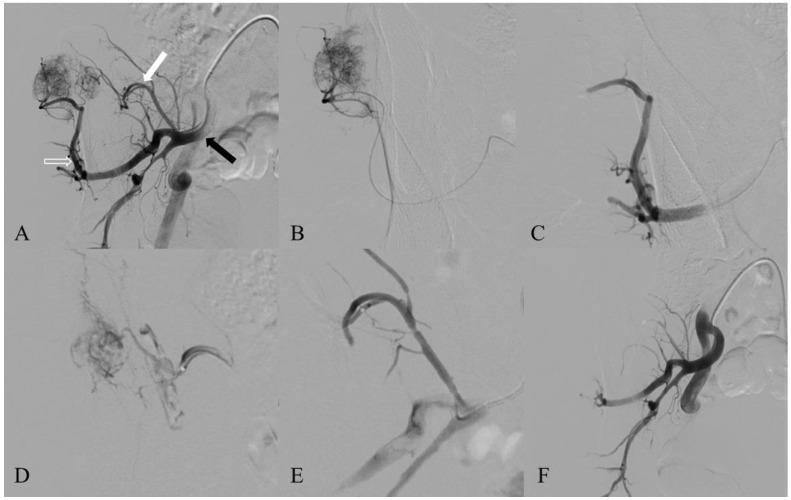
A fifty-eight-year-old man treated for right ilium metastasis of a clear cell renal cell carcinoma. (**A**) First angiographic acquisition, which was realized thanks to a 5-Fr catheter, showed a hypervascularized tumor fed by a main artery, namely, the posterior trunk of the right intern iliac artery (black arrow). Two feeders were identified: one arising from the superior gluteal artery (open arrow) and another arising from the iliolumbar artery (white arrow). (**B**) The feeder arising from the superior gluteal artery was catheterized selectively using a microcatheter Maestro^®^ 2.8 F (Merit Medical, South Jourdan, UT, USA) and embolized with 300–500 µm microspheres (Embospheres^®^, Merit Medical, South Jourdan, UT, USA). (**C**) The angiographic control showed a complete occlusion of that feeder. (**D**) The feeder arising from the iliolumbar artery was catheterized selectively using a microcatheter Maestro^®^ 2.8 F (Merit Medical, South Jourdan, UT, USA) and embolized with 300–500 µm microspheres (Embospheres^®^, Merit Medical, South Jourdan, UT, USA). (**E**) Control angiogram shows complete occlusion of that feeder. (**F**) Final angiographic control showing complete devascularization of the right ilium tumor, with a tumoral blush intensity reduction greater than 90%.

**Table 1 cancers-15-02657-t001:** Baseline characteristics of the 31 patients who underwent preoperative arterial embolization of bone tumors.

Characteristics	Values
Age (years)	67 (55; 74) [19–87]
Sex	
Male	15/31 (48%)
Female	16/31 (52%)
Tumor histological type	
Primary tumors	4/31 (13%)
Aneurysmal bone cyst	2/31 (7%)
Giant cells tumor	1/31 (3%)
Undifferentiated spindle and pleomorphic sarcoma	1/31 (3%)
Secondary tumors	27/31 (87%)
Clear cell renal cell carcinoma	17/31 (55%)
Chromophobe renal cell carcinoma	2/31 (7%)
Mucinous tubular and spindle cell carcinoma	1/31 (3%)
Solitary fibrous tumor	3/31 (10%)
Leiomyosarcoma	1/31 (3%)
Thyroid carcinoma	3/31 (10%)
Tumor localization	
Pelvic bone	6/31 (19%)
Acetabulum	5/31 (16%)
Ilium	1/31 (3%)
Thigh	12/31 (39%)
Femoral head	2/31 (7%)
Femoral Neck	1/31 (3%)
Femoral metaphysis	1/31 (3%)
Femoral diaphysis	5/31 (16%)
Femoral condyle	1/31 (3%)
Soft tissue	2/31 (7%)
Upper limb	10/31 (32%)
Humeral neck	2/31 (7%)
Humeral metaphysis	1/31 (3%)
Humeral diaphysis	6/31 (19%)
Forearm soft tissue	1/31 (3%)
Spine (sacrum)	3/31 (10%)

Qualitative variables are expressed as proportions; numbers in parentheses are percentages. Quantitative variables are expressed as medians; numbers in parentheses are interquartile ranges (q1; q3); numbers in square brackets are ranges.

**Table 2 cancers-15-02657-t002:** Anatomical characteristics of 84 embolized feeding arteries during 31 procedures.

Artery	Value
**Abdominal aorta**	1/84 (1%)
*Medial sacral artery*	1/84 (1%)
**Internal iliac artery**	**22/84 (26%)**
*Anterior trunk*	3/84 (4%)
*Pudendal artery*	1/84 (1%)
*Obturator artery*	3/84 (4%)
*Posterior trunk*	1/84 (1%)
*Trunk of superior gluteal artery*	3/84 (4%)
*Acetabular branch of superior gluteal artery*	1/84 (1%)
*Iliolumbar artery*	4/84 (5%)
*Trunk of lateral sacral artery*	4/84 (5%)
*Superior branch of lateral sacral artery*	1/84 (1%)
*Inferior branch of lateral sacral artery*	1/84 (1%)
**External iliac artery**	**3/84 (4%)**
*Inferior epigastric artery*	2/84 (2%)
*Deep circumflex iliac artery*	1/84 (1%)
**Common femoral artery**	**1/84 (1%)**
*Innominate branch*	1/84 (1%)
**Deep femoral artery**	**21/84 (25%)**
*Medical circumflex branch*	7/84 (8%)
*Lateral circumflex branch*	8/84 (9%)
*Muscular branch*	1/84 (1%)
*Perforating branch*	5/84 (6%)
**Superficial femoral artery**	**7/84 (8%)**
*Descending genicular artery*	2/84 (2%)
*Perforating branch*	5/84 (6%)
**Popliteal artery**	**3/84 (4%)**
*Superior lateral genicular artery*	2/84 (2%)
*Superior medial genicular artery*	1/84 (1%)
**Axillary artery**	**14/84 (17%)**
*Subscapular artery*	1/84 (1%)
*Anterior humeral circumflex artery*	2/84 (2%)
*Posterior humeral circumflex artery*	12/84 (14%)
**Humeral artery**	**13/84 (15%)**
*Innominate branch*	3/84 (4%)
*Branch from deep humeral artery*	6/84 (7%)
*Deep humeral artery*	4/84 (5%)
**Radial artery**	**1/84 (1%)**
*Radial recurrent artery*	1/84 (1%)
**Ulnar artery**	**1/84 (1%)**
*Interosseous branch*	1/84 (1%)

Variables are expressed as proportions; numbers in parentheses are percentages. Arteries in bold are the main large- or medium-caliber arteries from which the embolized feeders arised. Arteries in italics are the feeders (i.e., arterial branches vascularizing the tumor in which embolic agents were injected).

**Table 3 cancers-15-02657-t003:** Surgical data and outcomes of 31 patients who underwent preoperative arterial embolization of bone tumors.

Variable	Value
Type of surgery	
Resection alone	5/31 (16%)
Resection with prosthesis	8/31 (26%)
Resection with osteosynthesis	2/31 (7%)
Resection with both prosthesis and osteosynthesis	1/31 (3%)
Curettage alone	2/31 (7%)
Curettage with prosthesis	1/31 (3%)
Curettage with osteosynthesis	2/31 (7%)
Prosthesis alone	4/31 (13%)
Osteosynthesis alone	6/31 (19%)
Surgical intent	
Curative	4/31 (13%)
Palliative	27/31 (87%)
Hemoglobin, blood loss and need for transfusion	
Hemoglobin serum level before surgery (g/dL)	11.8 (10.9; 13.3) [9.2–15.2]
Hemoglobin serum level after surgery (g/dL)	10.5 (9.6; 11.3) [8.4–12.7]
Hemoglobin differential (g/dL)	1.9 (0.9; 2.7) [0.1–6.2]
Blood loss (mL)	300 (300; 1000) [300–1000]
Need for transfusion	9/31 (29%)
Packed red blood cells	3 (2; 4) [1–4]
Complications	2/31 (7%)
Functional results	
Complete improvement	8/30 (27%)
Partial satisfying improvement	15/30 (50%)
Partial unsatisfying improvement	4/30 (13%)
Lack of improvement	3/30 (10%)
Local recurrence	6/31 (19%)
Global disease progression	15/31 (48%)
Death	5/31 (16%)
OS (months)	22.1 (5.1; 29.7) [1.4–40.7]

Notes. Qualitative variables are expressed as proportions; numbers in parentheses are percentages. Quantitative variables are expressed as medians; numbers in parentheses are interquartile ranges (q1; q3); numbers in square brackets are ranges. OS indicates overall survival.

## Data Availability

The data that support the findings of this study are available from the corresponding author (M.B.) upon reasonable request.

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
