# Peer review of "Preoperative Arterial Embolization of Musculoskeletal Tumors: A Tertiary Center Experience"

_cancers, 2023, doi:10.3390/cancers15092657_

Round 1
Reviewer 1 Report
This is a well written paper presenting retrospectively collected results from preoperative embolization of bone and muscular tumors, primary or metastatic. The paper is educational and informative for clinicians and interventional radiologists. I have no suggestion for change.
Reviewer 2 Report
I thank the editorial board of Cancers for the opportunity to review this manuscript.
This work is devoted to the topical and not fully studied issue of mechanical and chemical transarterial embolization (TAE) in the treatment of musculoskeletal tumors of primary and secondary genesis. The results of this study have significant scientific and practical significance. It is clear that the authors of the manuscript know who, when and how to perform TAE. This is an additional factor that their opinion should be listened to and the results of their work should be considered.
The retrospective nature of the study, the small number of patients and the heterogeneity of diseases are limitations of the study. Nevertheless, it is a high-quality scientific work, and the statistical analysis presented confirms the validity of the results.
I have no fundamental comments.
Below are comments that will improve the quality of the manuscript and help readers better understand the results of the study.
1. Page 3, line 96-97: «… when important intraoperative bleeding was anticipated.»
Could the authors clarify when they expected major (substantial –слово из статьи. В переводе –значительная кровопотеря) blood loss during surgery?
2. Page 4, line 144-145: «Tumor-feeding artery occlusion was made using different embolic agents, whose selection was left at the discretion of the operator in the absence of robust recommendations»
Maybe the operator had some guidelines, based on his own experience, to choose some embolic agents or combination of embolic agents: diameter of the main feeding vessel, tumor size, primary or secondary tumor? Can you specify them? The authors point out that there are no precise recommendations on the choice of embolic agents, their opinion can be taken into account when making such recommendations.
3. Page 5, line 203: «However, they assessed evolution of pain…»
It was probably a visual analogue scale, maybe you should be presented а decrease in post-treatment pain in points? Quantitative assessment is always more convincing.
4. Page 10, line 325-326 «Among the nine patients who required blood transfusion, four (44%) had near-complete devascularization on control angiogram»
After which surgical procedures was blood transfusion required? Was it probably related to the volume of surgery, the size and localization of the tumor? Were they palliative or radical surgeries? Please be more specific.
5. Page 11, line 369-371 «However coils are not recommended as the sole embolic agent in tumor embolization because they cannot provide distal vessels embolization, which is necessary for hypervascular tumors with multiple collaterals»
Could the authors formulate what they consider to be the optimal combinations of embolic agents and the indicators/factors that guide their choice of embolic agents combinations? According to the authors, what is the optimal ratio when Glubran dilution with Lipiod?
6. Write a subsection on «Limitations». This will certainly strengthen the position of the manuscript. The authors point out the limitations of the study in the Discussion (page 12, lines 388-393), but these are only part of the limitations.
7. Flow chart of the study should be complemented, because it does not reflect all aspects of the study. Or it should be removed altogether, as it does not carry any additional information in this form.
